

# *AnnotationBustR*: an R package to extract subsequences from GenBank annotations

Samuel R. Borstein and Brian C. O'Meara

Department of Ecology & Evolutionary Biology, University of Tennessee, Knoxville, TN, USA

## ABSTRACT

**Background:** DNA sequences are pivotal for a wide array of research in biology. Large sequence databases, like GenBank, provide an amazing resource to utilize DNA sequences for large scale analyses. However, many sequence records on GenBank contain more than one gene or are portions of genomes. Inconsistencies in the way genes are annotated and the numerous synonyms a single gene may be listed under provide major challenges for extracting large numbers of subsequences for comparative analysis across taxa. At present, there is no easy way to extract portions from many GenBank accessions based on annotations where gene names may vary extensively.

**Results:** The R package *AnnotationBustR* allows users to extract sequences based on GenBank annotations through the ACNUC retrieval system given search terms of gene synonyms and accession numbers. *AnnotationBustR* extracts subsequences of interest and then writes them to a FASTA file for users to employ in their research endeavors.

**Conclusion:** FASTA files of extracted subsequences and accession tables generated by *AnnotationBustR* allow users to quickly find and extract subsequences from GenBank accessions. These sequences can then be incorporated in various analyses, like the construction of phylogenies to test a wide range of ecological and evolutionary hypotheses.

## INTRODUCTION

The use of DNA sequence data is vital for a wide variety of research in evolutionary biology and ecology. Molecular phylogenies, which rely on DNA sequences for their construction, are extremely prevalent in biological research. Whether being used to correct for shared ancestry among organisms (*Felsenstein, 1985*), or to test hypotheses related phylogeography (*Avise et al., 1987*), diversification (*Hey, 1992*; *Maddison, 2006*), and trait evolution (*Hansen, 1997*; *Bollback, 2006*), phylogenies are required. Additionally, the use of phylogenies is important in community ecology to place systems into an evolutionary framework (*Webb et al., 2002*; *Cavender-Bares et al., 2009*). The construction of molecular phylogenies for systematic purposes is also a popular tool for taxonomists to identify new taxa and classify organisms (*De Queiroz & Gauthier, 1994*; *Tautz et al., 2003*). Some DNA sequences, like the mitochondrial gene cytochrome oxidase subunit I

Corresponding author
Samuel R. Borstein,
sborstei@vols.utk.edu

(COI), have utility as a method to identify and catalog species using DNA barcoding (*Hebert et al., 2003*; *Ratnasingham & Hebert, 2007*, *2013*).

Sequence databases like GenBank provide a valuable resource for using DNA sequence data to test evolutionary and ecological hypotheses. With the reduction in cost of DNA sequencing and the advancement of methods to analyze sequence data, the amount of sequence data available for use is growing at a rapid pace. Given that GenBank has over one-trillion sequences from over 370,000 species (*Benson et al., 2017*) and recent advances in methods to create massive phylogenies using either super-matrix (*Driskell et al., 2004*; *Ciccarelli et al., 2006*) or mega-phylogeny approaches (*Smith, Beaulieu & Donoghue, 2009*; *Izquierdo-Carrasco et al., 2014*), the ability to generate large DNA sequence data sets for comparative analyses has become fairly common (*Leslie et al., 2012*; *Rabosky et al., 2013*; *Spriggs, Christin & Edwards, 2014*; *Zanne et al., 2014*; *Shi & Rabosky, 2015*). Additionally, sequence retrieval with command line utilities like National Center for Biotechnology Information's eutils (*NCBI Resource Coordinators, 2017*) as well as within common scripting environments for biological analyses, like R (*R Development Core Team, 2018*), Perl (*Perl Development Team, 2017*), and Python (*Python Software Foundation, 2016*), are made possible with packages like *ape* (*Paradis, Claude & Strimmer, 2004*), *rentrez* (*Winter, 2016*), *reutils* (*Schofl, 2015*), *seqinr* (*Charif & Lobry, 2007*), *Bioperl* (*Stajich et al., 2002*), and *Biopython* (*Chapman & Chang, 2000*).

While GenBank provides a wealth of sequence data for researchers to use, some of it is rather difficult to manipulate into a useful form. For example, some sequences may be concatenated together, or the only gene sequence available for a species for the locus of interest may be within a mitochondrial or chloroplast genome. At the time of writing, GenBank has 70,048 complete or partial mitochondrial genomes, and 4,698 chloroplast genomes; and a simple search for sequences containing concatenated coding sequences and tRNAs resulted in 286,538 additional sequences exclusive of the previously mentioned sequences. Although GenBank's annotation system provides a means to see where a locus of interest is in a genome or concatenated sequence and provides the ability to download it manually, this is extremely time-consuming when many accessions are involved, and not a feasible way to extract mass amounts of sequence data for use in research. While alternative sequence databases exist, especially for popular loci utilized for DNA barcoding and microbial community identification (ex. COI, 16S, etc.), these databases typically are extremely focused on just a few loci (*DeSantis et al., 2006*; *Ratnasingham & Hebert, 2007*; *Cole et al., 2013*). While these databases typically house data from GenBank, they may not have complete overlap with sequence data on GenBank, such as those from complete organelle sequences (ex. GenBank KR150862.1 for the mitochondrial genome of *Bujurquina mariae*, which is a species not included in the Barcode of Life Database even though the GenBank accession has a record for COI).

Another major challenge to obtaining large amounts of sequence data is the highly variable nomenclature of gene names. Most genes have several alternative names and symbols that are present in sequence databases. Among distant taxa, it is common for homologous genes to vary considerably in nomenclature (*Tuason et al., 2003*). Even within a group of closely related taxa or within a single taxon itself, how genes are annotated may

**Table 1 Functions and data included in the package *AnnotationBustR*.**

| Function/Data name | Description |
| --- | --- |
| AnnotationBust | Writes found subsequences for loci of interest to a FASTA file for a vector of GenBank accessions and writes a corresponding accession table. |
| data(cpDNAterms) | Loads a data frame of search terms for chloroplast genes. |
| data(mtDNAterms) | Loads a data frame of search terms for animal mitochondrial genes. |
| data (mtDNAtermsPlants) | Loads a data frame of search terms for plant mitochondrial genes. |
| data(rDNAterms) | Loads a data frame of search terms for ribosomal DNA genes and spacers. |
| FindLongestSeq | Finds the longest sequence for each species in a set of GenBank accession numbers. |
| MergeSearchTerms | Merges two or more data frames containing search terms of features to extract into a single data frame. |

differ substantially from record to record and a wide variety of alternative gene names may be found for a single gene (*Morgan et al., 2004*; *Fundel & Zimmer, 2006*). This poses serious problems when searching through databases for molecular sequence data (*Mitchell, McCray & Bodenreider, 2003*; *Tamames & Valencia, 2006*).

Here, we present the R package *AnnotationBustR* to address the issues discussed above. *AnnotationBustR* reads GenBank annotations in R and pulls out the gene(s) of interest given a set of search terms and a vector of taxon accession numbers supplied by the user. It then writes the sequence for the gene(s) of interest to FASTA formatted files for each locus that users can then use in further analyses. For a more in-depth introduction to using *AnnotationBustR* users should consult the vignette in R through `vignette("AnnotationBustR-vignette")`, which provides instructions on how to use the different functions and their respective options. Other details about the package can be accessed through the documentation via `help ("AnnotationBustR")`.

## DESCRIPTION

*AnnotationBustR* is written in R (*R Development Core Team, 2018*), a popular language for analyzing biological data, and requires R version 3.4 or higher. It uses the existing R packages *ape* (*Paradis, Claude & Strimmer, 2004*) and *seqinr* (*Charif & Lobry, 2007*). *AnnotationBustR* uses *seqinr*'s interface to the online ACNUC database to extract gene regions of interest from concatenated gene sequences or genomes (*Gouy et al., 1985*; *Gouy & Delmotte, 2008*). ACNUC is a database and retrieval system for molecular sequence data maintained by the Pôle Bio-Informatique Lyonnais, which provides access to GenBank data and allows for easy access and manipulation of complex sequences, such as trans-spliced genes that may be on opposite strands of DNA. A list of the currently implemented commands is given in Table 1 and a flow chart of function usage is shown in Fig. 1.

The main function of *AnnotationBustR*, `AnnotationBust`, takes a vector of accession numbers and a data frame of synonym search terms to extract loci of interest and writes them to a FASTA formatted file. This function also returns an accession table of all the loci of interest and the corresponding accession numbers the loci were extracted from for each
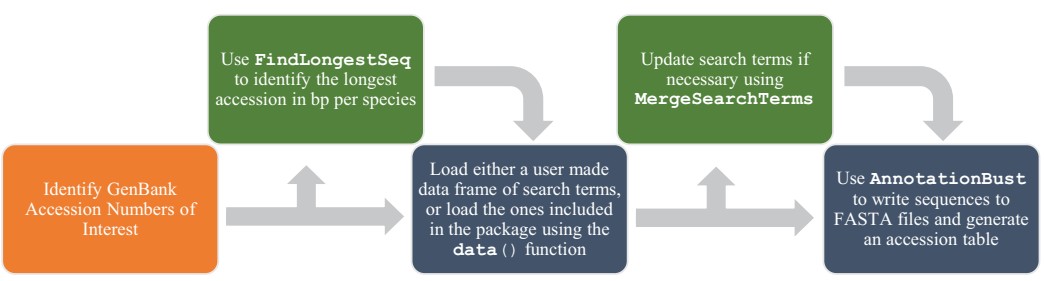

**Figure 1** **Flow chart of functions for a complete usage of *AnnotationBustR*.** Blue boxes indicate a step using the package *AnnotationBustR* while orange boxes represent steps that need to be completed outside of *AnnotationBustR*. Boxes in green represent optional steps in the AnnotationBustR pipeline.

species that can then be written to a csv file. Users can specify that duplicate genes should be extracted as well. If extracting coding sequences, users can also specify if they would like to translate the sequence into the corresponding peptides by specifying the appropriate GenBank numerical translation code.

We have included pre-made data frames with search terms in *AnnotationBustR* for animal and plant mitochondrial genomes, chloroplast genomes, and rDNA. These can be used to easily extract DNA barcodes, like COI for animals in mitochondrial genomes (*Hebert et al., 2003*), the internal transcribed spacers (ITS) in rDNA for fungi and plants (*Kress et al., 2005*; *Schoch et al., 2012*), and maturase K and ribulose-bisphosphate carboxylase genes in the chloroplast genome of plants (*Hollingsworth et al., 2009*). These pre-made data frames consist of three columns with the column `Locus` containing the output file name, `Type` containing the type of sequence it is (i.e., CDS, tRNA, rRNA, misc_RNA, D-loop, etc.), and the third column, `Name`, containing a possible synonym of the loci to search for. For example, for COI, GenBank includes gene names of COI, CO1, COX1, cox1, COXI, cytochrome c oxidase subunit I, and COX-I. An additional column, IntronExonNumber, is used to specify the intron or exon number for extracting introns and exons. These search terms can be loaded into the workspace using the `data()` function. Annotation files for each accession are read in through *seqinr* and regular expressions matching of the synonyms provided by the user to the feature annotations are performed to identify the subsequence to extract. As certain loci may have numerous synonymous listings in GenBank feature tables that may not be included in the pre-made data frames of search terms, *AnnotationBustR* has the function `MergeSearchTerms` which allows users to add additional search terms to a pre-existing data frame of search terms if users follow the basic column formatting stated above. An additional feature of *AnnotationBustR* is the function `FindLongestSeq` which finds the longest sequence for each species in a set of GenBank accessions.

To demonstrate the performance of *AnnotationBustR,* we timed how long it took to extract 13 popular coding sequences from 100 chloroplast genomes, the 13 coding sequences from 100 metazoan mitochondrial genomes, and the three ribosomal RNA genes and ITS 1 and 2 from 100 metazoan rDNA sequences (Fig. 2, see code in Data S1). Timings were performed for each accession number starting with the extraction of a single

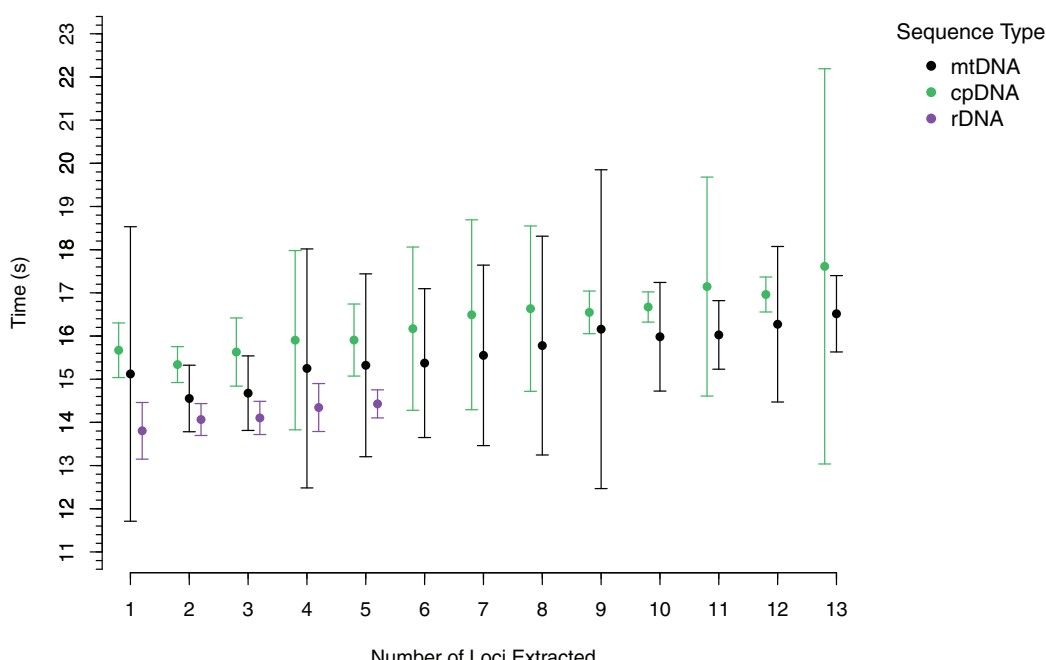

**Figure 2** Timings of subsequence extraction using `AnnotationBust` for 13 metazoan mitochondrial coding sequences (black), 13 chloroplast coding sequences (green), and five metazoan rDNA subsequence (purple). Points represent the mean time in seconds with bars representing +/− one standard deviation.

locus and progressively adding an additional locus until all targeted loci were extracted for that accession. Timing trials were performed on a Windows desktop with a 3.8 GHz Intel Core i7 processor and 64 GB of RAM. For each accession, we timed how long it took to extract one through the full number of subsequences sought. Our timings indicate that *AnnotationBustR* can efficiently extract these loci into FASTA files and that performance scales well as the number of loci to extract increases. Our timings also scale well in terms of sequence input size, as there is over variation in mean input sequence size between the rDNA (1,140 bp), mtDNA (16,560 bp), and cpDNA (149,971 bp) sequences. From profiling the code, it appears that the variation in the mean times to extract sequences is related to speed of the ACNUC server in returning responses to requests. While we did perform timing trials on a machine with a decent amount of RAM, *AnnotationBustR* is not very taxing on memory. The peak RAM usage of the `AnnotationBust` function during extraction for the largest dataset, the chloroplast genomes (average size of 149,971 bp), was only 87.35 megabytes when profiled using the R package peakRAM (*Quinn, 2017*).

*AnnotationBustR* is available through CRAN (https://cran.r-project.org/package=AnnotationBustR) and is developed on GitHub (https://github.com/sborstein/AnnotationBustR). New extensions in development and fixes can be seen under the issues section on the packages GitHub page. Active lists of synonyms to be implemented in the next version are available on our github site in the issues section, as we gather more from users. While ACNUC currently only provides access to RefSeq virus sequences from the

RefSeq database, we plan to add compatibility in the future for RefSeq accessions if access to other RefSeq databases is made available in *seqinr*.

## EXAMPLES

We provide an example of the utility of sequences obtained using *AnnotationBustR* relative to single sequences on GenBank in North American leuciscine minnows. This group was chosen as they have been widely sequenced, both for whole mitochondrial genomes as well as single mitochondrial genes. Specifically, we extracted 12S rRNA, 16S rRNA, COI, CYTB, and ND2 genes from 60 species of leuciscine minnows and one outgroup, the zebrafish (*Danio rerio*). These five genes were chosen as individual gene sequences have also been regularly sequenced in this clade for phylogenetic studies (*Bufalino & Mayden, 2010*; *April et al., 2011*; *Schoenhuth et al., 2012*). We used PHLAWD v. 3.3 (*Smith, Beaulieu & Donoghue, 2009*) to identify GenBank accessions to download for single genes that had the best coverage and identity for each species. PHLAWD uses BLAST (*Altschul et al., 1990*) restricted to a specified clade of interest to identify the best sequence in terms of coverage and identity for each species relative to a set of known bait sequences. We specified sequences had to have at minimum of at least 20 percent identity and coverage to be kept in the dataset of sequences accessed from GenBank on April 16, 2018. Multiple sequence alignment of sequences obtained from mitochondrial genomes using *AnnotationBustR* and of individual GenBank genes was performed using MAFFT v. 7.402 (*Katoh & Standley, 2013*). Phylogenetic reconstruction and bootstrap analysis was performed in RAxML v. 8.2 under the GTRGAMMA model of sequence evolution with the alignment partitioned by gene (*Stamatakis, 2014*) (Data S1).

The alignment constructed using sequences extracted from mitochondrial genomes using *AnnotationBustR* was 6,537 bp and slightly longer than the 6,508 bp alignment constructed from individual GenBank sequences. The total number of sequences was 305 (extracted from the 61 mitogenomes) and 198, in the *AnnotationBustR* and GenBank alignments respectively. The most notable difference between the two alignments is the amount of missing data. The GenBank alignment was only 37.72% complete relative to the 97.59% complete *AnnotationBustR* alignment. The disparity in number of sequences and completeness of alignments is due to some species only having sequence data for a locus within a mitochondrial genome or other concatenated sequence and not as a single sequence on GenBank. While we do recover slightly different topologies between the datasets, we find that the phylogeny reconstructed from the *AnnotationBustR* extracted sequences has higher bootstrap support among shared bipartitions relative to the tree constructed from individual GenBank sequences, with shared bipartitions amongst the phylogenies having an average of 93.69% and 81.56% bootstrap support in the *AnnotationBustR* and GenBank trees respectively (Fig. 3). Both tree topologies are similar to those recovered in other phylogenetic studies of leuciscine minnows (*Bufalino & Mayden, 2010*; *Hollingsworth et al., 2013*; *Martin & Bonett, 2015*). Our results may not necessarily reflect any aspect of quality of sequence data used, but rather quantity. For example, all the sequences extracted using *AnnotationBustR* are the complete sequence

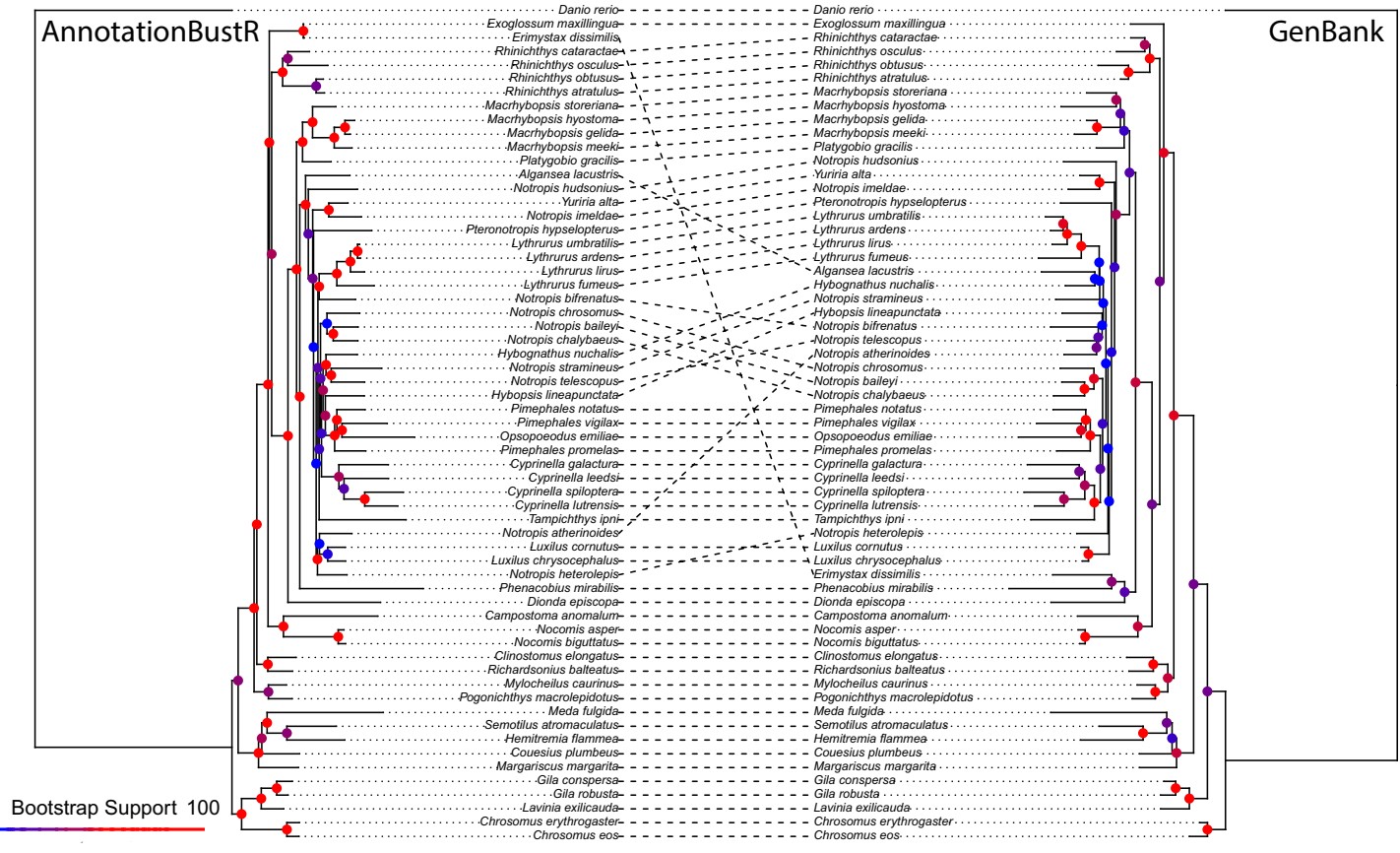

**Figure 3 Phylogeny of 60 North American leuciscine minnows from sequences extracted using *AnnotationBustR* and from individual GenBank sequences.** Circles at nodes represent bootstrap support, with warmer colors representing higher bootstrap support and cooler colors representing lower bootstrap support.

for that gene. This is not necessarily true of the single gene sequences obtained from GenBank, where some species only have a partial sequence for a gene. Additionally, some species only have sequence data for certain genes as a mitochondrial genome accession and not necessarily have any other accession for that gene.

## CONCLUSION

*AnnotationBustR* provides a way for users to extract subsequences from concatenated sequences, plastid, and mitochondrial genomes where gene names for subsequences may vary substantially. The major limitation to the functionality of *AnnotationBustR* is that it is only as good as the annotations in the features table it is using for extraction. For instance, some concatenated sequences do not have the individual gene positions annotated for the record and just state that it contains the genes, therefore making it impossible to extract a gene from it (ex. GenBank KM260685.1. GenBank KT216295.1). Additionally, some loci may be present in the sequence yet missing from the features table completely (ex. mitochondrial D-loop missing in GenBank KU308536.1). Another limitation is

that some popular loci such as intergenic spacers and some introns are not always annotated in the features table, making them impossible to extract. A good example of this is the trnH–psbA intergenic spacer, a proposed locus for plant DNA barcodes (*Kress et al., 2005*). As *AnnotationBustR* will extract all sequences within the annotations it finds that fit the supplied search terms, it is not a guarantee that these sequences are properly annotated, as annotation errors in sequence data are a known problem (*Ben-Shitrit et al., 2012*). Users will have to use due diligence on the sequences to check for incorrect annotations, mislabeled taxa, paralogs, chimeric sequences, and so forth. Alternative methods for extracting concatenated sequences could involve aligning them to known bait sequences of interest using a program like BLAST (*Altschul et al., 1990*). This can be automated by writing scripts to use BLAST locally or through NCBI. This represents a different approach from ours, relying on sequence similarity rather than annotations (PHLAWD builds on this but adds additional checks). While this can be useful, it potentially could be problematic as sequences could potentially align to non-orthologous sequences, which may cause issues with downstream analyses (*Lassmann & Sonnhammer, 2005*). At large phylogenetic scales, bait sequences may blast to paralogous copies rather than to orthologous sequences of more distantly related taxa in the clade. Nonetheless, BLAST and tools building on it represent historically useful approaches for creating a matrix of sequences that may be quite complementary to use of *AnnotationBustR* as another quality control check in the curation of large molecular sequence datasets.

While eutils and packages in popular programming language can provide access to sequence data, and in some cases access to subsequences, they require complex query language. The R package we have developed provides a simple way for users to extract subsequences that have variable annotations by supplying a vector of accessions and search terms, either the ones included in the package or their own curated set. While R may not be as widely used as Python or Perl as a bioinformatics platform, it is a widely popular scripting language and we feel that this package fills a need in the R community.

## CITATION

Researchers publishing a paper that has used *AnnotationBustR* should cite this article and indicate the version of the package they are using. Package citation information can be obtained using citation ("AnnotationBustR").

## ACKNOWLEDGEMENTS

We thank members of the O'Meara lab, Cedric Landerer, and Christopher Peterson for helpful discussions while developing the package and Orlando Schwery and Frankie West for beta testing. We thank the *seqinr* package authors for technical help related to their package. We thank three anonymous reviewers for their helpful feedback on improvements for the paper and the software.

### Funding
This work has been supported via a GTA to Samuel R. Borstein by the University of Tennessee, Knoxville. The funders had no role in study design, data collection and analysis, decision to publish, or preparation of the manuscript.

### Grant Disclosures
The following grant information was disclosed by the authors:
University of Tennessee, Knoxville.

### Competing Interests
The authors declare that they have no competing interests.

### Author Contributions
- Samuel R. Borstein conceived and designed the experiments, performed the experiments, analyzed the data, contributed reagents/materials/analysis tools, prepared figures and/or tables, authored or reviewed drafts of the paper, approved the final draft.
- Brian C. O'Meara authored or reviewed drafts of the paper, approved the final draft.

### Data Availability
CRAN: https://cran.r-project.org/package=AnnotationBustR
GitHub: https://github.com/sborstein/AnnotationBustR

### Supplemental Information
Supplemental information for this article can be found online at http://dx.doi.org/10.7717/peerj.5179#supplemental-information.

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
