# Peer review of "AnnotationBustR: an R package to extract subsequences from GenBank annotations"

_PeerJ, doi:10.7717/peerj.5179_

## Round 0.1 · original submission · Major Revisions

The three reviewers all identified strengths and weaknesses of the study. Reviewer 1 has serious doubts that the work will be an important contribution to the field, and I agree with many of the issues raised here. However, because reviewers 2 and 3 are less concerned with the overall impact and significance to the community, I am willing to extend the offer to substantially revise the manuscript. While all of the criticisms of the reviewers should be adequately addressed in your revision, the following issues should be especially addressed.

A thorough evaluation of AnnotationBustR should be performed, comparing the strengths and novelty of its functions to exiting approaches (please see Reviewer 1 for concerns RE existing functionality at NCBI that is not discussed).

More information needs to be provided on the exact types of data that can be utilized with the program. Also, because the algorithm is annotation-dependent, how are erroneously annotated sequences handled using AnnotationBustR? There are many, many annotation problems on GenBank and other databases, so this is not a minor problem to simply dismiss.

If one of the main reasons to employ AnnotationBustR is to circumvent errors with sequence similarity-based searches using annotation-based retrieval, it is crucial to show an improvement over the well-established approach using sequence similarity. One can firmly argue that sequence similarity is a vital factor in determining the robustness of annotations across a large group of sequences.
In general, many specifics RE the practicality and efficiency of AnnotationBustR are not provided. Please see the concerns raised by all three reviewers, but especially Reviewer 1. Consider making some of the comparisons, and especially consider utilizing eUtils.
Please consider the various audiences that will not only read this work, but wish to utilize the program. Is it practical for population genetics? Large-scale phylogeny estimations? Phylogenomics studies? Also, is it practical to use? Please understand that two of the three reviewers had difficulty operating the program, and that in order for your work to be published, the scripts should be easily downloaded and operational, with clear and consider descriptions for usability.

Please address Reviewer 2's concern for how AnnotationBustR will remain current and efficient provided that sequence repositories are an endless thriving tower of babel. Also concerning is how multiple different locus tags/accession number formats will hold up to the search criteria implemented with AnnotationBustR.

While the reviewers (and myself) appreciated the vignette, please consider making it clearer for the many concerns raised above.

Reviewer 1 ·

Basic reporting

The manuscript is grammatically correct, contains adequate background material to place the current work in the appropriate context, and is liberally referenced. The raw code is readily available on github, and easily installed into an existing R environment. Including a tutorial is a valuable addition. The figures are clearly labeled and described.

There are several instances of awkwardly long sentences that take away from the clarity of the writing; for example, lines 13-16, 61-63, 74-75, 125-129. More importantly, while the manuscript contains a substantial number of references, the literature used to support a central tenet of the study – that a major challenge to obtaining large amounts of sequence data is the highly variable nomenclature of gene names – is 10-15 years old. The field of bioinformatics has progressed dramatically since that time; for example, GenBank now includes synonyms in gene symbol searches, numerous data-specific repositories have arisen to provide gold-standard sequence data for barcoding genes like rDNA, many sites have fairly simple APIs for data access, etc.

Experimental design

The authors have developed an R package to extract data (particularly sequences) from GenBank using annotation text and synonym matching. They argue that this is potentially advantageous over sequence similarity (e.g., BLAST) for downstream analyses, particularly the construction of phylogenies, and allows easier access to underlying sequence data itself.
It is not clear that the current software is a particularly meaningful addition to the field. First, no mention is made of GenBank’s own eUtils, which provides straightforward access to sequence data using url strings. At least a comparison should be presented between eUtils and AnnotationBustR with respect to speed, usability, and accuracy. Second, no data is presented showing how much sequence data in GenBank is present in concatenated sequences versus discrete GenBank entries. Third, annotation errors in sequence data are a significant and ongoing problem, for reasons that are fairly well known but difficult to address (for one example of many, see Ben-Shitrit et al. 2012 Nature Methods 9: 373-8). It is not clear how AnnotationBustR accounts for these errors, which could insert substantial errors into downstream analyses. Fourth, the authors present no data (historical or in the current study) indicating that reliance on sequence similarity is a significant contributor to erroneous phylogenetic analyses, or that text matching such as that employed in AnnotationBustR would show any improvement over existing methods. This may, in fact, be the case; however, sequence similarity has been firmly entrenched in tree building for many years, and it is incumbent upon the authors to provide data to support using annotation strings instead.

Validity of the findings

The benchmark comparisons in Figure 3 are interesting – I expect the large variation around some data points is due primarily to differences in network latency – but I would have enjoyed more discussion around hardware requirements. For example, which is more important to AnnotationBustR performance: processor count, RAM, or network speed? Also, it is not clear how many times each test was performed.

Importantly, there is no analysis of AnnotationBustR’s accuracy or sensitivity. For example, do I retrieve fewer false positives using AnnotationBustR vs. BLAST? What about eUtils? How do results and usability compare between the current software and gold-standard databases like RDP or Greengenes?

In general, while this software may prove to be useful, the appropriate comparisons and benchmarks have not been included here. Without these data, the relevance of this contribution is not clear.

Additional comments

Please include hardware and software requirements for AnnotationBustR in the manuscript. I discovered during review that it does not appear to function properly on R version 3.3 (OS X) but does on R version 3.4. Also, running the AnnotationBust step of the tutorial throws several errors of the form:

Warning messages:
1: In if (grepl("unknown accession", full.rec) == TRUE) { :
the condition has length > 1 and only the first element will be used

·

Basic reporting

Because this manuscript describes a software package, I will be reviewing not only the manuscript but also the code and documentation provided.

I find that all data is shared appropriately. In addition to completely open source code, the distribution of a curated set of synonyms for organellar and ribosomal DNA sequences will be very valuable for the community. I was able to execute the example code used to generate the timing figure, with similar results on my Macbook Pro.

ANUNC is used several times in the introduction and abstract without a definition. I had to look it up-- its definition (as a sequence retrieval system) should be explained.

The number of additional synonyms is likely to proliferate as GenBank grows. Do the developers have plans to keep the cpDNA, mtDNA, and rDNA tables updated with the package?

The authors should mention that the mtDNA list they have provided with the package is intended for animal mitochondrial genomes. Many of the loci in the Brassica oleracea mitochondrial reference sequence (JF920286.1) are not shown, including several of the ribosomal proteins (rpl5, rpl16) and the tag arginine translocation gene (tatC). Although mitochondrial coding regions evolve slowly in plants, they are frequently used for deep phylogenetics. See, for instance: Liu, Y., Cox, C. J., Wang, W., & Goffinet, B. (2014). Mitochondrial Phylogenomics of Early Land Plants: Mitigating the Effects of Saturation, Compositional Heterogeneity, and Codon-Usage Bias. Systematic Biology, 63(6), 862–878. doi:10.1093/sysbio/syu049

Experimental design

Here I will report on conceptual issues with the AnnotationBustR package, its utility, and suggestions for minor improvements to the package and text.

Why don't refseq accessions like NC_ work? Is it possible to extract the GenBank accession number given a refseq accession number?

Why are intron sequences not supported? In plants, intron sequences within plastid genes are often quite useful for phylogenetics (for example, within rps2 or matK). Could the second column of the annotations feature column be used to extract the intron from a named gene while also using the synonym utility of AnnotationBustR?

It would be useful if the vignette gave an example of retrieving sequences with different IDs in GenBank. I did not go through all of the mtDNA examples, but the four great ape sequences chosen for cytochrome odixase were all identified by COX1. If there already is such an example, it should be highlighted in the vignette as well as within the manuscript text.

Because the package uses seqinr as a dependency, it would also be useful to see a full example employing a workflow that includes searching for gene sequences (the orange box in Figure 1). For example, one use of AnnotationBustR would be to retrieve all trnG sequences from the plant genus Oenothera. Some GenBank sequences from this genus are deposited as individual genes, while others are full chloroplasts. How would I manipulate a GenBank search using seqinr to fully utilize AnnotationBustR? I realize that GenBank search results will have to be manually curated, but adding this example to the workflow and tutorial would probably enhance the usability of the package.

Validity of the findings

Here I will report some issues and suggestions I found when running the R vignette or code contained in help documentation.

While the vignette is formatted fine within RStudio, it is not formatted properly in Github, due to a recent change in Git Flavored Markdown-- a space is required between the ## and the header. The images also do not appear on GitHub.

In the vignette, under troubleshooting, fix this sentence:

"if you have internet access yet are still haivng isues, check that these sites and there servers are not down"

The help documentation for AnnotationBust says that the function returns a data.frame containing a list of sequences that “can be turned into an accession table using MakeAccessionTable”. I cannot find any other reference to this function.

As is, the data.frame currently combines multiple accessions from each species into a comma-delimited table, which may not be easily used in other formats. In the tutorial example, the two human sequences are joined into a single row. Why not have one row for each sequence source?

It should be noted in the documentation that the default behavior is to overwrite existing files when code is re-run. Users should be notified/warned that this is occurring. Users could also benefit from an option to append to existing files.


When I run AnnotationBust for either the R vignette tutorial code or my own example, I always get the following errors:

When the code is first executed:
In socketConnection(host = host, port = port, server = server, blocking = blocking, :
pbil.univ-lyon1.fr:5558 cannot be opened

Upon code completion:
Warning messages:
1: In if (grepl("unknown ", full.rec) == TRUE) { :
the condition has length > 1 and only the first element will be used

The FASTA files all began with a newline character. This will break some FASTA parsers which rely on detecting the > as the first character.

Why are the nucleotides returned in lower case? The sequences are not stored in GenBank this way. Some sequence parsers will treat these sequences as soft masked. Perhaps give the user an option about case?

Additional comments

I approached this review as a potential user of the software package, and as someone who works primarily in phylogenetics of plants. The scripts included with AnnotationBustR will be useful for researchers who wish to extract reasonably well-curated protein coding sequences from GenBank. The suggestions I have made will hopefully improve the usability and documentation of the R package.

Reviewer 3 ·

Basic reporting

Borstein and O'Meara present a package that provides important functionality: extracting subsequences from GenBank based on the annotations associated with sequences. While there are a few tools that allow for the extraction of features from GenBank files, none of these are particularly easy to, for example, incorporate into scripts and automated workflows.

In general, the manuscript is well written. However, I found the introduction to be written with a strange audience in mind. For example, shouldn't it be obvious that molecular sequence data is vital for biological research these days. Not sure if that needs to be stated. And is it necessary to say that molecular phylogenies are "prevalent" in biological research? If the reader doesn't know that, then they probably don't know what a molecular phylogeny is. I would recommend rewording some of this for a more consistent audience (e.g., choose whether the reader knows phylogenies / molecular work or not). Then again, perhaps I am off base and this better reflects an audience group that I am simply overlooking.

Otherwise, the basic reporting and writing in the manuscript is well done.

Experimental design

I would mainly question the use of R for this task. R is not traditionally used as an informatics platform for many reasons. R is not particularly fast for text and is somewhat more obtuse for incorporating into informatics pipelines. I wonder if the authors might be able to write a paragraph that describes why this would be beneficial. I am not suggesting that they authors write this in another language, just wondering why R was chosen.

I am also a little surprised that other methods or other packages aren't discussed. For example, do the functions presented here have more functionality or are they better than the SeqRecord class functions in BioPython? Also, while I am less familiar with it, there surely must be similar functions in BioPerl. This may be addressed by responding to the comment mentioned above, but is there a reason that these are not adequate for the authors use? I see no mention of either of these widely used packages in the manuscript. If the functionality isn't there, then it would seem like this fills a gap. If the functionality is there, it doesn't render your R package less useful, but it would be nice to understand the similarities and differences (i.e., perhaps it is the same and it is extending this functionality to R which is growing as an informatics platform?).

I appreciate the posting of the code on GitHub and the vignette describing how it may be used.

Validity of the findings

The conclusions presented in the paper are well founded and the limitations are well stated. I would only suggest discussing the intersection of this work with other informatics platforms. Again, not to express that one is better than another or any of that, but instead so that users may better understand what functionality is present and not.

---

## Round 0.2 · Major Revisions

Dear Drs. Borstein and O'Meara:

Thank you for submitting your revision. Unfortunately, I was only able to receive one review from the original reviewers. This reviewer acknowledged that you have added significant text content to the manuscript, and implemented a new feature in your software to discriminate between intronic and exonic data. However, the reviewer asserts that there remains a complete lack of analysis of the validity of the data being generated by your tool. I agree with this assessment. This is a crucial aspect of any bioinformatic pipeline described in similar reports of this nature, and I feel strongly that you should carry this out in order for your manuscript to be published. I hope that you choose to do so, and I would be happy to consider such a revision for publication in PeerJ.

Good luck.

Reviewer 1 ·

Basic reporting

The awkward sentences have been improved where noted, which improves the readability.

Experimental design

The authors have developed a usable software tool for extracting genomic regions of interest from GenBank records in an automated fashion. This is a fundamental task in bioinformatics that underlies many different areas of modern biology. The software presented here relies on a text matching approach, rather than the more traditional use of sequence similarity. This approach has its merits, which are presented well in the manuscript, but it also includes several serious, well-documented challenges, including spelling differences, errors in annotation, annotation of true orthologs as hypothetical or unknown, generic annotation strings that may not indicate true function, and more. AnnotationBustR uses a synonym lookup table to address spelling and syntax differences, but does not address the subtler challenges. Admittedly, much of this may be beyond the scope of the current work; however, as discussed in more detail below, this manuscript would benefit from some analysis that estimates the impact of those issues on the data being produced by the described software.

Validity of the findings

The software is functional and appears to be fairly straightforward to maintain, assuming continued access to the ACNUC database. It does have the advantage of a relatively simple interface, increasing its audience of potential users, and the choice of platform is appropriate – although some might argue that R has a significant learning curve, the same can be argued of virtually any language. Of more concern is assessing the validity of the data being generated by AnnotationBustR, particularly vis-à-vis phylogeny estimation where high quality data is absolutely required. Since this is a stated motivation for developing AnnotationBustR, the current manuscript would benefit greatly from some effort to benchmark the success of this approach. This is particularly true given the myriad issues with gene annotation, some of which are discussed by the authors, upon which AnnotationBustR relies. For example, one might use a double-blind approach, assembling data sets independently using AnnotationBustR and another method or two and comparing the output. Or it might be valuable to provide a comparison between an AnnotationBustR-informed phylogeny and an existing published tree based on the same loci. While AnnotationBustR may be easy to use, it is incumbent upon the authors to assess the quality of the data that it generates so as not to invalidate their hard work in crafting a usable tool.

---

## Round 0.3 · Minor Revisions

Dear Drs. Borstein and O'Meara:

I have received a re-reiew from one original reviewer of your work. While I invited the other two original reviewers to look at your revision, they havent replied.

I agree with the concerns of the reviewer, and I anticipate your work being ready for publication once these issues are addressed.

Best,
-joe

Reviewer 1 ·

Basic reporting

The revised manuscript contains numerous changes to the text, and its clarity has improved. In the interest of reproducibility, please include software version numbers and dates of accession of public data sets, where applicable.

Experimental design

The authors have included a direct comparison between AnnotationBustR and an alternative sequence retrieval approach using PHLAWD. This is a notable addition and serves to strengthen the manuscript significantly. However, please describe how PHLAWD was used; specifically, what settings were employed to generate the alternative data set? This may explain the wide disparity between sequence coverage between the two approaches but it is not possible to evaluate. Also, why was PHLAWD employed instead of a simpler BLAST search using a target sequence and restricted by taxonomy? A reader would benefit from a brief explanation of the rationale behind the experimental design here.

Validity of the findings

While the benchmarking results shown in Figure 2 are interesting, they would be much more useful in comparison to existing methods. Alternatively, assessing the AnnotationBustR code itself (apart from its reliance on external tools) might prove more applicable to the current manuscript; for example, how do speed and memory requirements scale with respect to input size?

Additional comments

AnnotationBustR is a potentially useful tool for sequence retrieval, and a worthwhile addition to the R ecosystem. While the authors seem to bill their software as an alternative to sequence similarity (i.e. BLAST), I suggest that it would be quite useful as a companion tool - particularly given the level of manual inspection required for building a quality phylogeny regardless of the approach.

---

## Round 0.4 · accepted · Accept

Dear Drs. Borstein and O'Meara:

Thanks for further revising your manuscript based on the minor concerns raised by the reviewer. I now believe that your manuscript is suitable for publication. Congratulations! I know it was a long process, but I firmly believe it was worth it. I look forward to seeing this work in print, and I anticipate it being an important resource for the informatics community. Thanks again for choosing PeerJ to publish such important work.

Best,

-joe

#